# Extracellular vesicles therapy alleviates cisplatin-ınduced testicular tissue toxicity in a rat model

Halime Tozak Yıldız[1]*, Kübra Tuğçe Kalkan[1], Numan Baydilli[2], Zeynep Burçin Gönen[3], Özge Cengiz Mat[4], Eda Köseoğlu[4], Gözde Özge Önder[4], Arzu Yay[4]

1 Department of Histology and Embryology, Faculty of Medicine, Kirsehir Ahi Evran University, Kirsehir, Turkey, 2 Department of Urology, Faculty of Medicine, Erciyes University, Kayseri, Turkey, 3 Betül-Ziya Eren Genome and Stem Cell Center (GENKOK), Erciyes University, Kayseri, Turkey, 4 Department of Histology and Embryology, Faculty of Medicine, Erciyes University, Kayseri, Turkey

* htyildiz@ahievran.edu.tr, hhalimeyildiz@hotmail.com

## Abstract

### Purpose

Cisplatin is a commonly used chemotherapy agent effective against various cancers, however it induces significant gonadotoxicity and infertility due to its adverse effects on testicular function. The underlying mechanisms of cisplatin-induced testicular damage include oxidative stress and dysregulated autophagy. This study investigates the potential of extracellular vesicles (EVs) to mitigate cisplatin-induced testicular damage through their regenerative, antioxidant, and autophagy-modulating properties.

### Methods

In the testicular toxicity model, thirty-two male rats were randomly divided into four groups (n = 8): control, EVs-only, Cis-only, and Cis + EVs. A single intraperitoneal dose of 7.5mg/kg cisplatin was administered on the first day. On the six day, the EVs treatment group received a single dose of EVs ($8 \times 10^7$/100µl) intravenously. Animals were sacrificed on day eight. Testicular histoarchitecture was assessed via hematoxylin and eosin staining. Sperm parameters, including motility and count, were measured using light microscopy. Hormone levels (testosterone and inhibin) were determined via enzyme-linked immunosorbent assay (ELISA). Oxidative stress markers, such as glutathione peroxidase (GSH-PX), superoxide dismutase (SOD), catalase (CAT), and is a metabolite malondialdehyde (MDA), were quantified using colorimetric assays. Autophagy and steroidogenesis were evaluated through immunohistochemical analysis of Beclin-1, p62, LC3–2, SF-1, and StAR.

**Data availability statement:** All relevant data for this study are publicly available from the figshare repository (https://doi.org/10.6084/m9.figshare.27214404).

**Funding:** The author(s) received no specific funding for this work.

**Competing interests:** The authors have declared that no competing interests exist.

## Results

Cisplatin exposure caused significant testicular damage, characterized by reduced germinal epithelium and degeneration of seminiferous tubules ($p < 0.001$). These structural changes led to hormonal imbalances, as evidenced by declines in testosterone ($p < 0.005$) and inhibin ($p < 0.001$). Additionally, sperm motility ($p < 0.05$) and count ($p < 0.001$) were adversely affected.

Immunohistochemical analysis revealed upregulation of autophagy markers ($p < 0.001$), indicating heightened autophagic activity, alongside downregulation of steroidogenic factors ($p < 0.001$), which contributed to impaired steroidogenesis. Elevated levels of malondialdehyde (MDA) ($p < 0.01$) and decreased activities of antioxidant enzymes—GSH-PX, SOD, and CAT ($p < 0.001$) pointed to increased oxidative stress as a contributing mechanism.

In contrast, treatment with extracellular vesicles (EVs) significantly improved testicular histoarchitecture ($p < 0.001$) and restored hormonal levels toward normal (testosterone $p < 0.005$, inhibin $p < 0.001$). Furthermore, EVs reduced the expression of autophagy markers ($p < 0.001$) and enhanced the levels of steroidogenic factors ($p < 0.05$). Notably, MDA levels decreased ($p < 0.001$), while antioxidant activities increased ($p < 0.001$), suggesting a protective effect of EVs against oxidative stress.

## Conclusion

EVs protect against cisplatin-induced reproductive toxicity by modulating oxidative stress and autophagy pathways, preserving testicular function and fertility. These findings suggest that EVs may be a promising therapeutic strategy for mitigating cisplatin's negative effects on reproductive health. Further exploration of dosing regimens and localized applications is recommended for improved efficacy.

## Introduction

Cisplatin (Cis) is a widely used chemotherapy drug for treating various types of cancer, including small cell lung, testicular, ovarian, and breast cancers. It functions primarily by forming DNA cross-links that inhibit DNA replication, leading to cell cycle arrest, apoptosis and autophagy in rapidly dividing cancer cells [1]. However, cis' use is limited by its significant side effects, including its toxicity to the reproductive system [2–10].

Cis' intracellular metabolites are primarily known as aquated metabolites (such as mono/di-aqua cis). Once inside the cell, cis undergoes hydrolysis, losing chloride ions and forming its active species [5]. The mechanism of cis-induced reproductive toxicity is closely associated with the production of reactive oxygen species (ROS) during its metabolism, which leads to oxidative stress and subsequent tissue damage [2,3,5,8–10]. In females, Cis has been shown to cause apoptosis in granulosa cells, which support the development of oocytes, leading to a reduction in ovarian reserve and impaired folliculogenesis [6]. In males, Cis negatively impacts testicular

function, affecting Leydig cells, seminiferous tubules, spermatogenic cells, and Sertoli cells, which are crucial for maintaining normal spermatogenesis [8]. This damage manifests as reduced sperm count, altered sperm morphology, impaired chromatin integrity, and decreased sperm motility, ultimately contributing to infertility [4–12]. The disruption of spermatogenesis by cis involves damage at multiple stages, including spermatogonia, spermatocytes, and spermatids, leading to impaired sperm development and maturation [12].

Extracellular vesicles (EVs) are a heterogeneous family of spherical, lipid-bilayered vesicles released from endosomal compartments or shed from the cell surface [13]. EVs contain a variety of bioactive molecules, such as proteins, lipids, mRNAs, miRNAs, tRNA, genomic DNA, cDNA, and mtDNA. As extracellular organelles, EVs play a crucial role in cellular information flow [14,15]. They initiate cell-to-cell communication by delivering their cargo of proteins, miRNAs, and enzymes to target cells, functioning as extracellular organelles in both the paracrine and endocrine systems. [13,16]. This category includes exosomes, microvesicles (MVs), and apoptotic bodies. EVs are present in extracellular fluids and various body fluids, including blood, urine, hydrothorax, saliva, breast milk, synovial fluid, and cerebrospinal fluid. EVs can be phagocytosed by other cell types, endocytosed, or directly fused with other cells [17]. EVs can reduce inflammation and promote tissue regeneration by transferring critical biomolecules such as proteins and RNAs from stem cells to injured cells. In cis-treated models, MSC-EVs have shown potential to alleviate oxidative stress and tissue damage, suggesting that they may serve as a promising alternative or complement to whole-cell MSC therapy in regenerative medicine [17].

Autophagy is a vital cellular process that maintains quality control by degrading and recycling intracellular components, including pathogens, persistent proteins, damaged macromolecules, and malfunctioning organelles [18]. This mechanism is essential for providing alternative intracellular substrates and energy, especially during nutrient deprivation, thereby supporting cell survival under stress [19]. Additionally, this process can be activated by various stressors such as infection, hypoxia, nutrient scarcity, and reactive oxygen species (ROS), functioning independently of other cell death pathways like apoptosis, thus underscoring its unique role in cellular homeostasis [19]. Notably, autophagy can be stimulated by anti-cancer drugs, which promote the degradation of tumor cells; however, while some oncoproteins inhibit autophagy, many tumor suppressor genes enhance it [20].

Cis, a commonly used chemotherapeutic agent, is known for its potent gonadotoxic effects, which manifest primarily through testicular damage, oxidative stress, and the activation of apoptotic pathways [3,4,11,12]. This damage affects key testicular cells, including spermatogenic cells, Sertoli cells, and Leydig cells, leading to disrupted spermatogenesis and reduced fertility [2,7]. Recent interest has focused on bone marrow-derived mesenchymal stem cell (MSC) extracellular vesicles (EVs), which have shown considerable promise in tissue regeneration, inflammation suppression, and immune modulation due to their diverse bioactive components [21,22]. While studies have explored the impact of various stem cells on testicular injury in Cis-induced experimental models, the therapeutic potential of EVs specifically remains under-explored. The therapeutic promise of EVs lies in their ability to deliver bioactive molecules that can modulate cellular pathways involved in damage and repair. Mechanistically, EVs have been shown to exert their effects by altering the expression of genes involved in oxidative stress, apoptosis, and autophagyand may provide a multifaceted approach to reduce drug-induced reproductive toxicity. This study aims to explore the regenerative potential of bone marrow-derived MSC EVs on Cis-induced testicular damage, focusing on their roles in modulating autophagy and oxidative stress pathways, which are key contributors to the observed gonadotoxic effects.

## Materials and methods

### Animals and experimental groups

The study was started with eight-week-old male Wistar albino rats (weight: 150–210 g). The rats were kept in a 25°C room with a 12-hour light/dark cycle, free access to water, and a standard diet from the Experimental and Clinic Research Center at Erciyes University in Kayseri, Turkey. After one weeks of acclimatization the rats were randomly divided into 4 groups, with 8 rats in each group.

The control group did not receive any form of treatment. The group treated with extracellular vesicles (EVs) received a single intravenous injection(i.v.) of EVs (8x10$^7$/100 μl) on the sixth day of the experiment [23]. The Cis group was administered a single intraperitoneal (i.p.) injection of cisplatin (7.5 mg/kg) (Kocak Pharma, Istanbul, Turkey) on the first day of the experiment to induce testicular toxicity [24,25]. In the Cis+Extracellular Vesicles (Cis + EVs) group, rats were administered a single i.p. injection of cisplatin (7.5 mg/kg) on the first day to induce cellular damage. On the sixth day, a single i.v. dose of EVs (8x10$^7$/100 μl) was administered to the rats. On the 8th day of the study, rats in the groups were sacrificed by cervical dislocation after testicular and blood samples were taken under xylazine (10 mg/kg, IP) and ketamine (60 mg/kg, IP) general anesthesia (Fig 1).

The experiments were conducted in accordance with The ARRIVE guidelines (Animal Research: Reporting of In Vivo Experiments). All experimental protocols were approved by the Local Ethic Committee of Animal Experiments of Erciyes University (approval no. 22/185)in Turkiye.

## Isolation of EVs from mesenchymal stem cells

Rat bone marrow derived mesenchymal stem cells were obtained from Erciyes University Genome and Stem Cell Institute (GENKÖK) in Turkiye. MSCs were thawed in a 37°C water bath and seeded in 75 cm² cell culture dishes (TP Inc., Rochester, NY, USA). When the density of MSCs reached above 90% confluency, serum-free medium (MEM-a, Cat no: BI01–042-1A; Biological Industries, Beit HaEmek, Israel) was added to each cell group and secretomes of the cells were collected at the end of 24 hours.

EVs isolation was performed from the obtained secretome using a standard commercial microvesicle quantitation kit ExoQuick-TC (#EXOTC50A-1, System Bio Sciences, Palo Alto, CA, USA) which is patented technology and already in the market [26]. The collected secretomes were centrifuged at 3000 x g for 15 minutes, as per the kit instructions. The supernatant was transferred to a new tube, and an exosome isolation solution was added at a ratio of 1:5 relative to the volume of the supernatant Vortex was used to assist homogenize it. After that, it was incubated for 24 hours at + 4°C. Following incubation, the mixture was centrifuged for 30 minutes at + 4 °C at 1500 x g. After centrifugation, the pellet was resuspended in PBS (Sigma-Aldrich, St. Louis, MO, USA), and the supernatant was discarded. The mean size of the secretome was measured using a Nanoparticle Tracking Analysis system (NTA, Malvern Instrument Nanosight NS300, Malvern, UK).

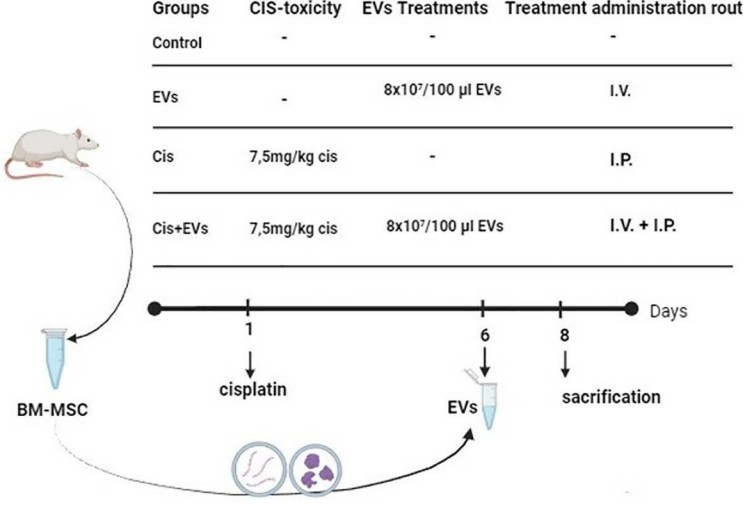

**Fig 1. Schematic representation of the experimental design.**

The particle live-imaging settings were set according to the manufacturer's software manual (NanoSight NS300 User Manual, MAN0541-01-TR-00, 2017) and the measurement was completed. As a result of the analysis using the nanoparticle device, secretomes with an average diameter of 81.6 nm were found (Fig 2). The Glomax Multi ELISA absorbance reader (Promega, Madison, WI, USA) was used to read absorbance values at 450 nm following the quantification process that followed the kit's instructions. The amount of EVs was calculated using standards [27]. EVs diluted at $8\times10^7/100\,\mu l$ were injected into the EVs and Cis + EVs groups of rats in the study.

## Assessment of sperm parameters

Epididymal sperm were collected from the cut edge of the caudal epididymis in 5 mL of 0.9% saline solution followed by gentle shaking on a rocker for ten minutes to achieve homogenization. The suspension was then incubated at room temperature for two minutes. For analysis, 10 µL of the supernatant containing epididymal sperm was diluted with %4 paraformaldehyde (Sigma-Aldrich,-Alabaster, AL, USA), (pH = 7.2–7.4) and citrate buffer (Bio-Optica Milano, Italia) to a total volume of 990 µL. Subsequently, approximately 10 µL of this diluted mixture was transferred to a Neubauer chamber (hemocytometer) and examined under light microscopy at 400x magnification. The pelleted cells on the chamber's surface were then counted. Sperm concentration was determined using the counted cells and the hemocytometer's dimensions, expressed in millions of sperm per mL, following Badkoobeh et al.

For sperm morphology assessment, around 20 µL of sperm suspension was mixed with an equal volume of eosin (Merck, Darmstadt, Germany) -nigrosin (Merck, Darmstadt, Germany) (1% eosin Y and 5% nigrosine). This staining method is based on the principle that viable cells exclude the dye, while non-viable cells absorb it. After, 2 mins incubation at room temperature, the slides were examined using an optical microscope (CX31, Olympus Optical of Brazil Ltda, São Paulo, Brazil) at 400x magnification. Dead sperms appeared pink and live sperms were not stained. For each sample, 200

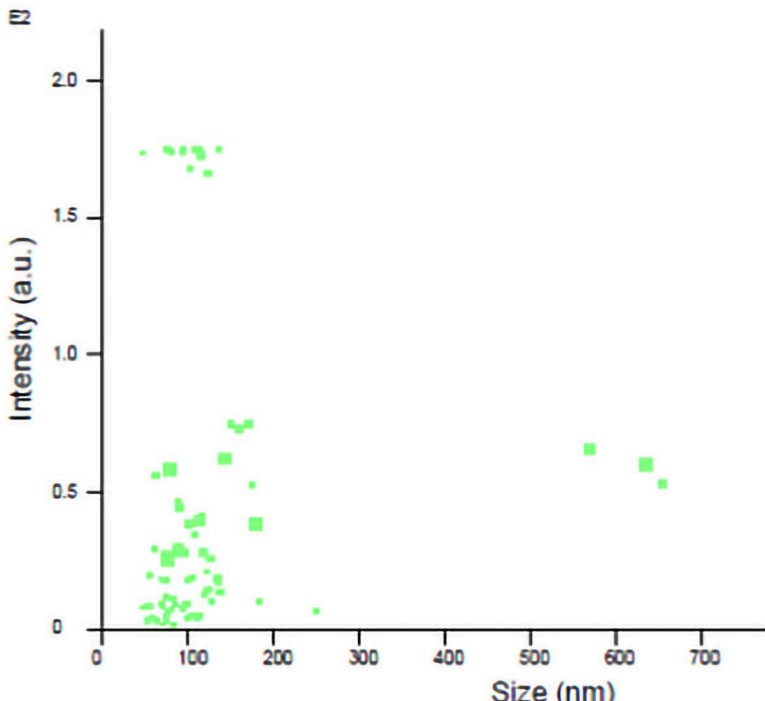

**Fig 2. Nanoparticle tracking analysis.**

spermatozoa were counted, noting changes in the head, middle piece, and tail. Results were expressed as the percentage of normal cells, following Badkoobeh et al. [28].

## Histopathological analysis

The left testis was bisected and fixed in 10% neutral buffered formalin (Sigma-Aldrich, St. Louis, MO, USA) for 48 hours. Following fixation, the tissues were subjected to a graded ethanol series (70%, 80%, 90%, 95%, and 100%) for dehydration and subsequently embedded in paraffin blocks for histological examination. Sections 5 μm thick were then cut from a paraffin block (Leica, RM 2000, Wetzlar, Germany) and stained with hematoxylin and eosin (H&E) (Thermo Scientific, Waltham, MA, USA) to examine the general histomorphological structure of testis tissues. Standard light microscopy was used to examine and evaluate the testicular tissue samples, with the analysis conducted under blinded conditions to ensure objectivity. The testicular tissue was scored according to Johnsen's criteria after being stained with H&E. [29] The Johnsen's testicular biopsy score (JBTS) was determined by sampling five locations in each section's 20 seminiferous tubules. According to the number of cells and how mature they were, JBTS was calculated for each tubule (Table 1). Moreover, as a measure of damage, the diameter of the seminiferous tubule was measured at a scale of 200 x using the ImageJ software (NIH, Bethesda, MD, USA.). The typical tubule diameter was determined by measuring the diameters of the seminiferous tubules at ten different locations.

## Immunohistochemistry

The streptavidin-biotin-peroxidase method was used for the immunohistochemistry staining kit (Lab Vision™ UltraVision™ Large Volume Detection System: anti-polyvalent, HRP, TA-125-HL, Thermo Fisher Scientific, Fremont, CA, USA.). Beclin-1, p62, LC3–2, steroidogenic factor-1 (SF-1), and steroidogenic acute regulatory protein (StAR) expressions in testicular tissues were identified using this technique. Testes were cut into 5-μm sections, deparaffinized in xylene, and then rehydrated using a graduated ethanol series. After rehydration, the sections were washed in deionized water, phosphate-buffered saline (PBS) was used to rinse the sections, and then 3% hydrogen peroxide (33% hydrogen peroxide solution was diluted with methanol) (Sigma-Aldrich, St. Louis, MO, USA) was applied for 5 minutes. All sections were then saturated in 5% sodium citrate buffer and heated at for 5 minutes in a microwave twice at $95^0C$ to initiate antigen retrieval. Beclin-1 primary antibody diluted 1:500 (Novus Biologicals, NB500–249, Littleton, CO, USA), p62 primary antibody diluted 1:200 (Anti-SQSTM-1, H00008878, Abnova Corporation, Taipei, Taiwan), LC3–2 primary antibody diluted 1:750 (Cell Signaling Technology, Danvers, MA, USA), SF-1 primary antibody diluted 1:150 (NBP2–46247; Novus Biologicals, Centennial, CO, USA), and StAR primary antibody diluted 1:250 (bs-3570R; Bioss Inc., Woburn, MA, USA) were all applied to the sections in an immunohistochemical chamber for an overnight duration at 4°C. Following that, the sections were incubated with biotinylated secondary antibodies for 10 minutes in a humid environment. After washing, the samples were incubated with 3,3'-diaminobenzidine tetrahydrochloride (10X DAB) substrate for 3 minutes at room temperature. (TA-060-HDX, Thermo Fisher Scientific, Waltham, MA, USA) to make the immunoreactivity visible (2.5mL of DAB with 22.5mL of

**Table 1. Johnsen testicular biopsy score.**

| Score | Histological findings | Score | Histological findings |
|---|---|---|---|
| 1 | No cells in the tubular section | 6 | There were few spermatids (5/tubule) |
| 2 | There were only Sertoli cells | 7 | There were too many spermatids without any sign of difference |
| 3 | Germ cells were just as spermatogonium | 8 | Late spermatids without mature spermatozoa |
| 4 | There were few spermatocytes (5/tubule) | 9 | There were few spermatozoa (5/tubule) |
| 5 | There were too many spermatocytes | 10 | Exact spermatogenesis was present with a large number of spermatozoa |

the Stable Peroxide Substrate Buffer). Gill's hematoxylin (Sigma-Aldrich, St. Louis, Missouri, USA) was used as a counterstain on the sections to enhance nuclear staining. Under a light microscope, each section was evaluated, and microscopic images of five different testicular tissue regions were obtained. Beclin-1, LC3–2, p62, SF-1, and StAR immunoreactivity intensities were measured using the Image J software (NIH; Washington, USA) [30].

### Enzyme-linked immunosorbent assay

The right testis was promptly bisected on an ice-cold mold, and blood samples were quickly centrifuged to separate the serum. Both the testis and serum samples were then placed into Eppendorf tubes and stored at -80°C for preservation. Tissues were transferred into PBS (0.01 M, pH 7.4) for homogenate preparation to be used in biochemical evaluations. Each homogenized sample was centrifuged at 4 ºC at 1500 x g for 10 min. For the ELISA method, the obtained supernatants were aliquoted. Glutathione Peroxidase (GSH-PX) (Sunred Bio, Cat. No. 201-11-5104; Shanghai, China), Superoxide Dismutase (SOD), Catalase (CAT) (Sunred Bio, Cat. No. 201-11-5106; Shanghai, China), and tumor necrosis factor-alpha (TNF-α) (Sunred Bio, Cat. No. 201-11-0765; Shanghai, China) were measured to determine the antioxidant levels of testicular tissue. Furthermore, Lipid peroxidation was assessed in terms of malondialdehyde concentration (MDA) (Sunred Bio, Cat. No. 201-11-0157; Shanghai, China) was measured in the testis tissue. The protein levels in tissue samples were determined using the Bradford method [31].. The levels of serum inhibine (INHB) (Sunred Bio, Cat. No. 201-11-0730; Shanghai, China) and testosterone (Sunred Bio, Cat. No. 201-11-0564; Shanghai, China) were assessed. A microplate reader was used to measure the wells' absorbance (Thermo Fisher Scientifc).

### Statistical analysis of the data

The GraphPad Prism version 9 was used for statistical analyses. The Shapiro-Wilk test was performed to determine if the data were normally distributed. Significant post-hoc comparisons of the variables were shown by the Dunn test for the Kruskal-Wallis analysis and the Bonferroni test for the one-way ANOVA test. The results of the P below 0.05 were regarded as statistically important for all data.

## Results

### EVs reduced the pathological changes in Cis-damaged testicular tissues

EVs reduced pathological changes in cis-damaged testicular tissues. Evaluation of testicular sections stained with H&E revealed that the control group's seminiferous tubules were organized into various stages of spermatogenesis and germinal epithelium, comprising spermatogonia, spermatids, and spermatocytes, with Leydig cells observed in the interstitial space. The Cis group showed degenerative alterations in the seminiferous tubules, including dilatation, reduced germ cell numbers, irregularities, and lumen shedding. In the EVs groups, the seminiferous tubules were regularly arranged, similar to the control group. Although edema was present in the Cis + EVs group, tubule degeneration was reduced, and the germinal epithelium appeared more regular. (Fig 3).

The experimental group's testicular injuries were scored using JTBS (Table 1). The Cis group scored significantly lower than the other groups. There is a significant difference between Cis and Cis + EVs (p < 0.001 for all comparisons) (Fig 4a).

Seminiferous tubule diameters were also evaluated to assess testicular injury. The Cis group had significantly smaller seminiferous tubule diameters than the control group and EVs group. Seminiferous tubule diameter increased statistically in the Cis + EVs group compared to the Cis group. (p < 0.05). There is a significant difference between the control, EVs, Cis (p < 0.001), and Cis + EVs groups (p < 0.05). (Fig 4b)

### The effect of EVs sperm count and motility

Following cis therapy, there was a significant decrease in sperm count compared to the control group (p < 0.001) and the EVs group (p < 0.001). A significant increase was observed between the Cis and Cis + EVs groups (p < 0.001) (Table 2).

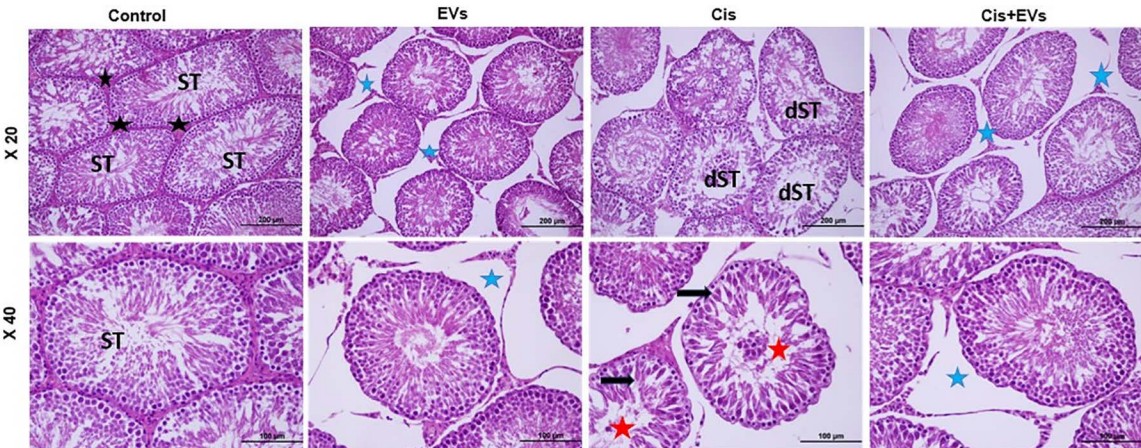

**Fig 3. Histology of testis sections (Olympus BX51, Tokyo, Japan X20: Scale bar; 200 μm, X40: Scale bar; 100 μm).** Effects of EVs on seminifer-ous tubules (ST) and interstitial tissue with Leydig cells (black asterisks) in testis tissues. Representative Hematoxylin and Eosin stained sections from indicated groups. Normal histology was observed in the control group. The Cis group experienced degenerative seminiferous tubule (dST) features. Black arrow; decreased number of germ cells, Red asterisks; lumen shedding, Blue asterisks; edema.

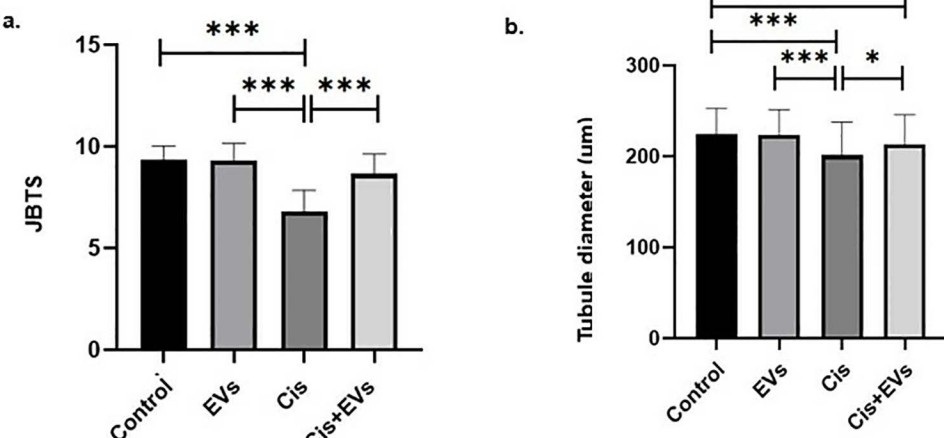

**Fig 4. JTBS and Tubule Diameters. a.** The general Johnsen's testicular biopsy scoring for all groups. **b.** Seminiferous tubule diameter measurement for all groups. *, p<0.05, **; p<0.01, **; p<0.001.

Sperm motility analysis showed a significant decrease post-Cis therapy when compared to both the control and EVs groups (p<0.05). Although the Cis + EVs group exhibited an increase in motility compared to the Cis group, the difference was not statistically significant. (Table 2).

### Cis-induced autophagy is supressed by EVs

This research investigated the contribution of MSC-derived EVs on the expression levels of autophagy proteins Beclin-1, p62, and LC3–2 following cis administration. In the control group, immunohistochemical evaluation of autophagy markers showed weakly positive cytoplasmic reactivity (Fig 5). The Cis group exhibited a substantial increase in immunoreactivity intensity for

**Table 2. Sperm count and motility.**

|  | Control (mean±SEM) | EVs (mean±SEM) | Cis (mean±SD) | Cis+EVs (mean±SD) |
|---|---|---|---|---|
| **Sperm Count (million/mL)** | 47.33±4.93 | 36.16±2.38$^{\$\$\$,\ ***}$ | 13.00±1.75*** | 16.00±1.09*** |
| **Sperm Motility (%)** | 51.66±4.77 | 50.83±8.20$^{\&\&}$ | 21.66±6.91** | 44.16±7.35 |

Effect of exosomes on sperm count (million/mL), and sperm motility (%) in testis tissues. ***. Control versus Cis, Control versus Cis+EVs, Control versus EVs, Cis+EVs versus Cis (p<0.001), $$$: Cis versus EVs (p<0.001), &&: Cis versus EVs (p<0.05), **: Control versus Cis (p<0.05).

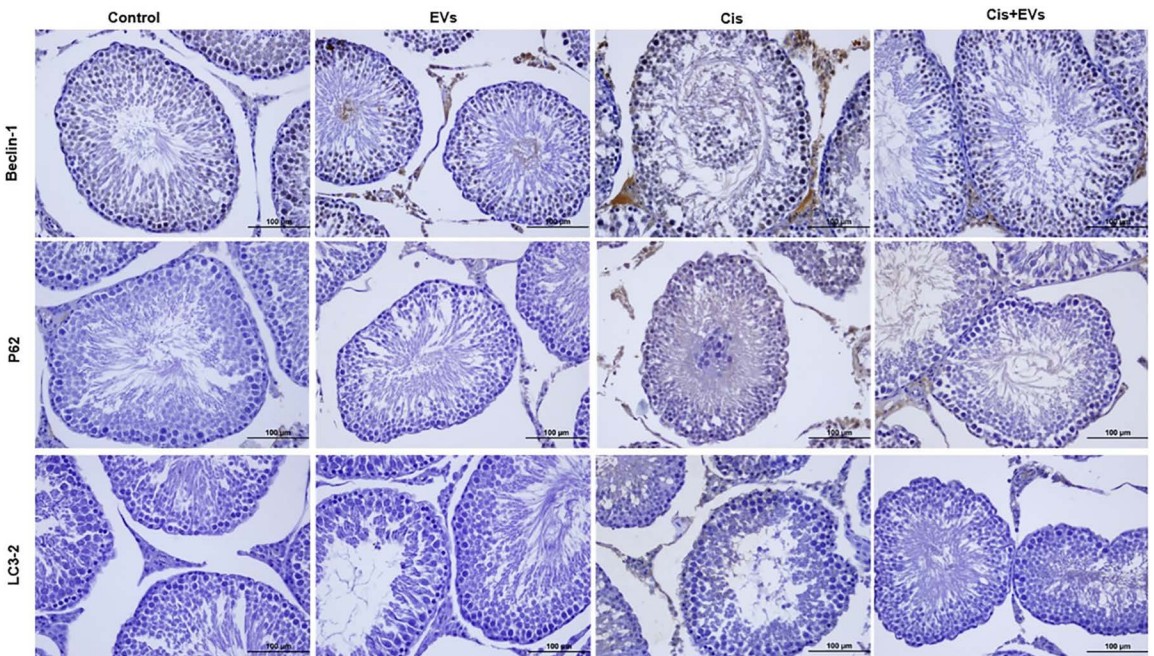

**Fig 5. Immunohistochemical analysis of testis sections (Olympus BX51, Tokyo, Japan. X40: Scale bar; 100 µm).** Levels of autophagy markers in the testes of control and treated group. EVs treatment ameliorated expression of Beclin-1, P62, and LC3-2 levels in the testes of cisplatin-induced rats. Avidin-biotin peroxidase technique.

Beclin-1, p62, and LC3–2 compared to the control group (p<0.001 for all comparisons). In EVs-treated rats, there was a significant decrease in the autophagy markers Beclin-1 (p<0.05), p62, and LC3–2 (p<0.001) following Cis treatment (Fig 6).

## SF-1 and StAR immunoreactivity

SF-1 has been shown to modulate the expression of the key protein StAR, which is crucial for testosterone synthesis. In the interstitial area, Leydig cells in the control group expressed SF-1 and StAR immunoreactivity (Fig 7a). The Cis group exhibited significantly lower immunoreactive intensity in Leydig cells compared to the control group (p<0.001). In the Cis+EVs group, rats treated with EVs after Cis administration showed a significant increase in SF-1 immunoreactivity in Leydig cells (p<0.05). However, StAR expression in the Cis+EVs group did not show a significant increase (Fig 7b).

## Effect of EVs on testicular oxidative stress and serum hormone levels

Testicular oxidative stress injury brought on by Cis was discovered by measuring antioxidant and oxidant enzyme levels in this study. When compared to the control group, testicular GSH-PX concentration decreased significantly (p<0.05),

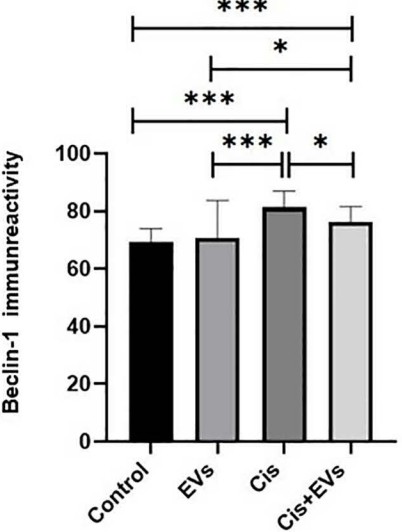
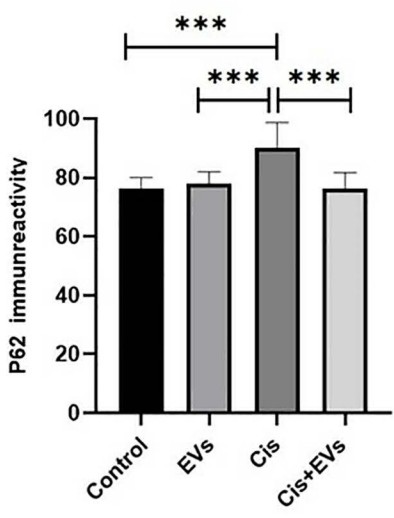
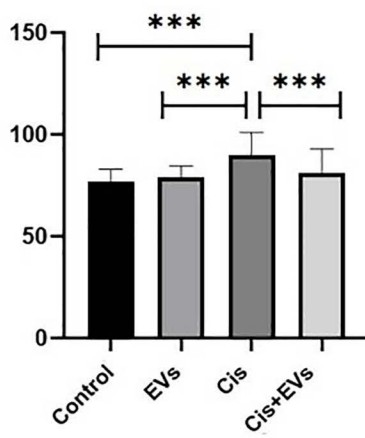

**Fig 6. İmmunoreactivity intensity of autophagy proteins.** Impact of EVs Beclin-1, p62, and LC3/2 in the testes tissue of rats treated with cisplatin. *,p<0.05, **p<0.01, ***p<0.001.

and oxidative stress was dramatically initiated, as indicated by an elevated MDA level (p<0.01). However, compared to controls, it decreased activity, SOD, and CAT (p<0.001). MDA production was reduced, and MDA imbalance was improved in the Cis + EVs group treated with EVs following Cis (p<0.001). EVs-therapy increased GSH-PX concentration in the Cis + EVs group rats (p<0.05) and significantly decreased oxidative stress by decreasing testicular MDA (p<0.001). When compared to the control group, there was a significant rise in testicular TNF-α expression in the Cis-treated group (p<0.05). The EVs-treated Cis + EVs group blocked the increase in testicular TNF-α expression compared to the Cis group, but there was no statistically significant difference (Fig. 8).

Reduced SF-1 and StAR immunoreactivity in Leydig cells resulted in decreased testosterone levels. Cis therapy decreased serum testosterone (p<0.005) and INHB levels (p<0.001) significantly (p<0.001). When compared to the Cis group, EVs therapy increased serum testosterone (p<0.005) and INHB levels (p<0.001) (Fig 9).

## Discussion

Cis, a widely used chemotherapeutic agent, is known for its effectiveness against various cancers; however, its use is associated with significant testicular toxicity due to its potent cytotoxic effects. Because chemotherapy agents target dividing cells, the spermatogenesis process is particularly affected. Previous studies have reported that exposure to Cis reduces reproductive function [2,9,10,12,32]. This study's histological, immunohistochemical, and biochemical results the damaging impact of Cis on testicular tissue.

Antitumor drugs that destroy cancer cells, such as Cis, damage tissue by inducing autophagy and apoptosis in cells [1–3,5,8–12,24,25]. Extracellular vesicles (EVs) derived from bone marrow mesenchymal stem cells (MSCs) have shown therapeutic potential in various studies on tissue damage and degenerative disorders [13,17,33–35]. EVs function as extracellular organelles with paracrine/endocrine roles, transporting proteins, miRNAs, and enzymes to target cells [17]. Many studies have reported that EVs have anti-apoptotic, anti-inflammatory, and regenerative effects [15,33–35].

Cis treatment adversely affects testicular morphology, triggers autophagy, reduces hormone levels and sperm count, and increased oxidative stress. A single dose of EVs therapy demonstrated significant regenerative and antioxidant

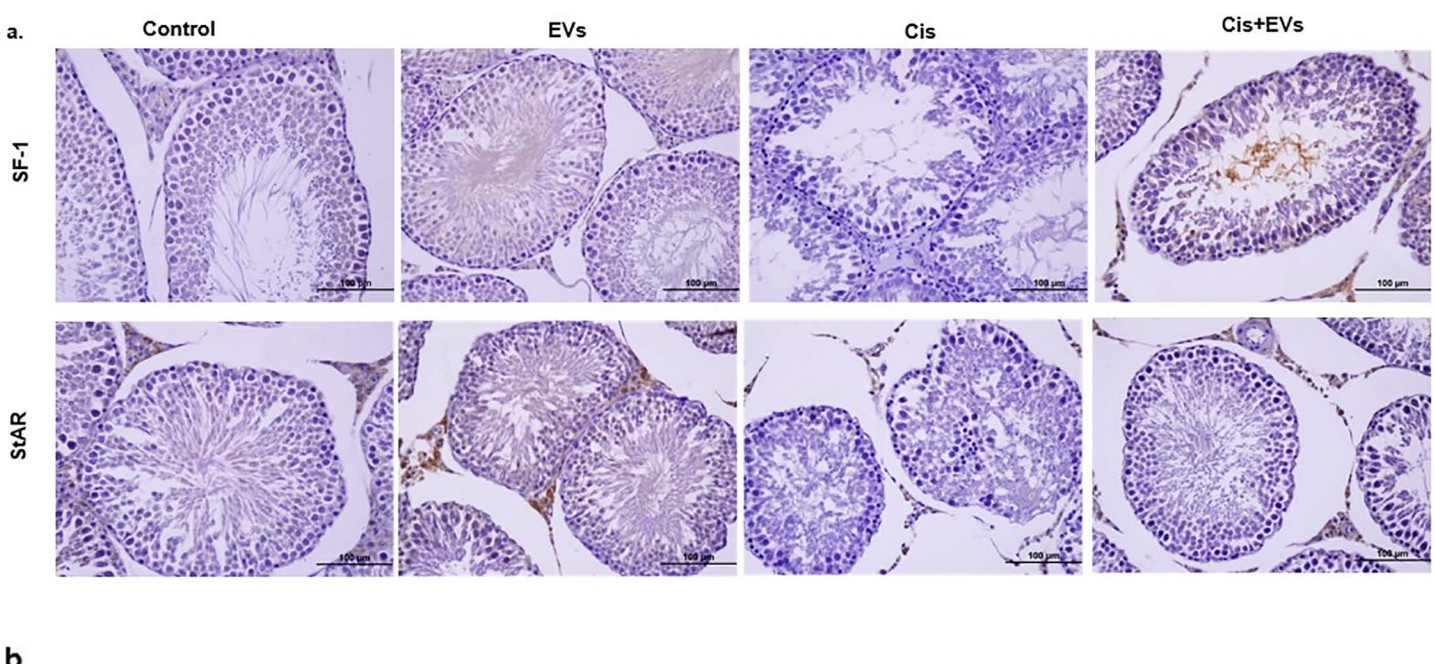

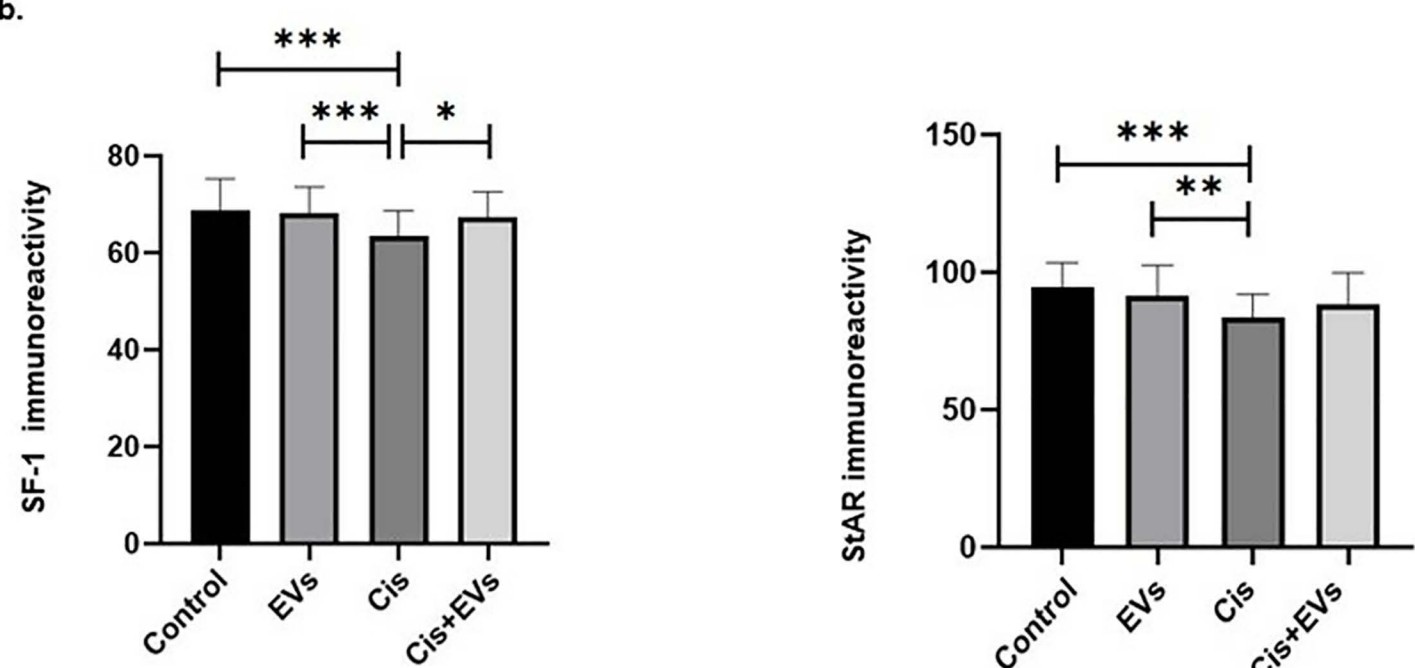

**Fig 7. Immunohistochemical analysis of SF-1 and StAR expression in interstitial area of testes (Olympus BX51, Tokyo, Japan. X40: Scale bar; 100 μm). a.** SF-1 and StAR expression in all experimental groups is shown in the illustrative figure. Avidin-biotin peroxidase technique. **b.** Immunoreactivity intensity of SF-1 and StAR in interstitial area of testes tissue. *, $p < 0.05$, **; $p < 0.01$, ***; $p < 0.001$.

effects, as well as regulation of autophagic activity. Despite the significant increase in hormone levels, the lack of a significant effect on sperm motility and count is due to the administration of EVs only two days prior to sacrifice, which is insufficient time for a a full effect on sperm previously stored in the epididymis. The Cis group exhibited degeneration, atrophied

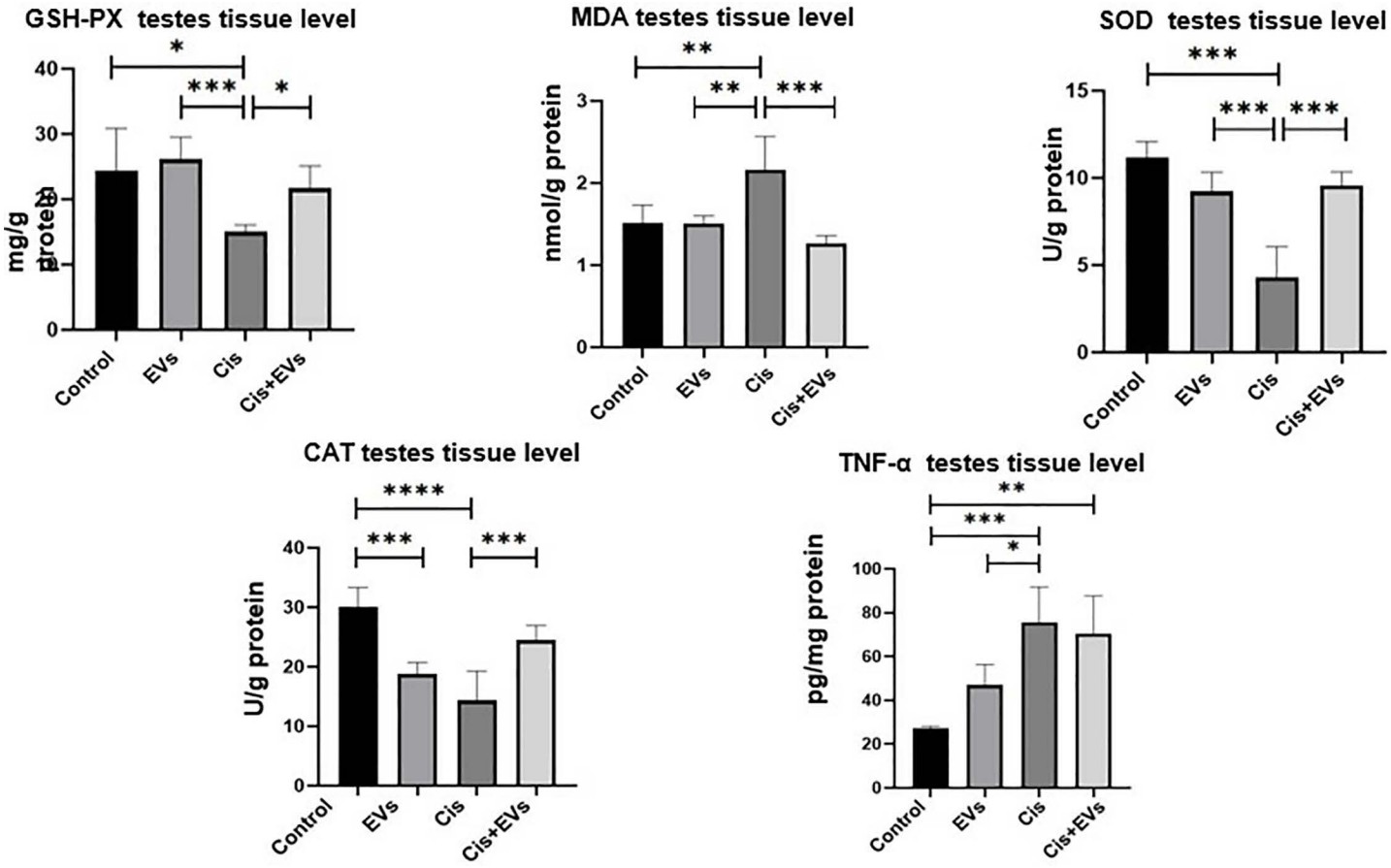

**Fig 8. Statistical analysis of testicular oxidative stress.** The effects EVs on oxidative stress parameters and apoptotic marker in cisplatin-induced testis damaged in rats. MDA, SOD, CAT, GSH-Px and TNF-α testes tissue level of the experimental groups.; *, p < 0.05, **; p < 0.01, ***; p < 0.001.

tubular epithelium, cavities, and dilation of seminiferous tubules. A reduction in tubule spermatogenic cells, germ cell infiltration into the lumen, and interstitial edema were observed. These findings were corroborated by a decrease in the JTBS score and tubule diameter. Previous studies reported that seminiferous tubule diameters, germinal epithelial thickness, and testosterone synthesis decreased in Cis-treated rats, potentially leading to testicular atrophy and infertility [36–38].

EVs are potent paracrine agents that play a crucial role in cellular communication and exhibit significant potential for tissue repair [21]. This study EVs treatment reduced seminiferous tubule degeneration and restored the structure of germinal epithelial cell. However, edema persisted in this group. Moreover, the JTBS score and tubule diameters increased in this group. They mediate cell-cell interactions by transporting proteins, miRNAs, and enzymes to target cells, acting as extracellular organelles with both paracrine and endocrine functions [16]. The proteins and miRNAs in EVs are involved in a variety of biochemical and physiological processes, such as cellular communication, structural maintenance, inflammation modulation, tissue repair and regeneration, and metabolic regulation [14,15,34].

The testicles have two primary functions: the production of testosterone and the generation of sperm [39]. Leydig cells, located in the interstitial tissue of the testes, are responsible for secreting testosterone [40]. In this study, the group treated with cis showed a reduction in testosterone levels, which was linked to decreased expression of SF-1 and StAR proteins. Research has shown that cis directly affects Leydig cells, resulting in cellular death and reduced secretory activity [25,41].

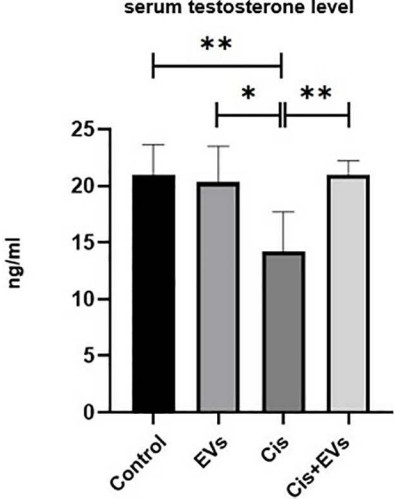
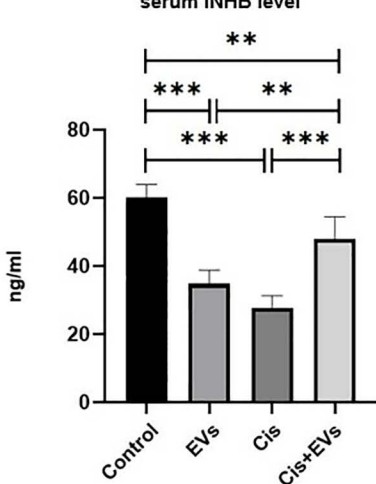

**Fig 9. Statistical analysis of serum hormone levels.** Serum testesteron and serum INHB level of the experimental groups.; *, p<0.05, **; p<0.01, ***; p<0.001.

INHB, a key hormone secreted by Sertoli cells, plays a crucial role in regulating reproductive function by providing negative feedback to follicle-stimulating hormone (FSH) [42,43]. Gonadotropin-releasing hormone (GnRH) released from the hypothalamus induces FSH in the pituitary (the hypothalamic-pituitary-testicular (HPT) axis), which stimulates the seminiferous tubules and Sertoli cells necessary for the initiation of spermatogenesis, a testosterone-dependent process [44]. In this study, the damaged seminiferous tubule epithelium and reduced testosterone levels resulted in significantly lower INHB levels in the cis-treated group compared to the control group. However, INHB levels notably increased in the group treated with EVs, likely due to tissue regeneration observed in the seminiferous tubules and interstitial regions. A decline in serum INHB is indicative of sperm damage and infertility, making it a valuable marker for assessing spermatogenesis [45,46]. Inhibins also function as growth factors in Leydig cells via paracrine signaling, indirectly supporting spermatogenesis alongside testosterone [47].

Cis, an alkylating agent, cross-links the inner and outer DNA chains, generating adduct forms that manifest as twisted DNA strands and DNA fragmentation which is a functional sperm parameter. High sperm DNA fragmentation (SDF) has been shown to negatively affect male fertilizing ability and pregnancy. This mechanism presumably affects sperm and testicular structure [36]. In this study the administration of cis on the first day of the study resulted in a notable reduction in sperm motility and count. However, by day six, the group that received EVs therapy (Cis+EVs) exhibited a significant recovery in sperm count, despite no substantial improvement in sperm motility. This suggests that while EVs therapy promotes spermatogenesis, it may not have an immediate impact on sperm motility during the observed timeframe. Cis damages Sertoli cells and weakens the blood-testis barrier, leading to DNA damage in germ cells at various developmental stages [48]. Cis effects on the testis are primarily driven by oxidative stress, apoptosis, and impaired cellular functions, which hinder the production of healthy sperm, leading to significant decreases in sperm count and motility, as shown in studies involving rat testes [24,37,38]. The compromised sperm, when stored in the epididymis, demonstrates a reduction in fertilisation capacity and motility due to the initial damage incurred. Consequently, the effects on epididymal sperm are secondary to the damage sustained during production in the testes. [24,49,50].

Tissue injury is often associated with acidosis, creating an acidic microenvironment. This acidic condition promotes the preferential uptake of EVs, particularly exosomes, by damaged cells via endocytosis. This mechanism highlights the potential of EVs in tissue-specific therapeutic interventions, as they naturally accumulate in areas of

tissue damage [15,51]. Cis-induced oxidative stress and tissue acidosis in testicular tissue stimulate both the apoptotic pathway and autophagy. İn this study autophagy induced by Cis in the testis is evidenced by increased levels of Beclin-1, LC3–2, and p62 proteins. During stress, in response to cellular damage, the immunolocalization of Beclin-1, LC3–2, and p62 proteins increases in the seminiferous tubules, particularly within Sertoli and germ cells. It has been reported that antitumor drugs can trigger the autophagy process [20]. A single dose of EVs therapy significantly improved tissue repair by regulating cis-induced autophagy. Autophagy, which is essential for cellular adaptability and survival, can be triggered by various stressors, including metabolic stress, endoplasmic reticulum (ER) stress, and chemotherapeutic drugs [52]. Spermatogenesis requires a significant amount of energy. During this process, factors such as starvation, chemical exposure, and radiation can have adverse effects. Studies have shown the connection between apoptotic and autophagic molecular mechanisms under these pathophysiological conditions [53,54]. BM-derived MSC exosomes protect against testicular ischemia/reperfusion injury in rats through antioxidant, anti-inflammatory, and anti-apoptotic mechanisms [54]. Adipose-derived MSC-EVs provide neuroprotection and enhance neurological recovery by inhibiting ischemia-induced autophagy [55]. EVs have demonstrated immunoregulatory, anti-inflammatory, regenerative, and anti-apoptotic capacities in therapeutic applications; however, whether these characteristics regulate autophagy remains unclear [17].

The therapeutic effects of EVs on cis-induced oxidative stress-induced testicular toxicity in male rats were evaluated in this study. The Cis group showed significantly lower GSH-PX, SOD, and CAT activities in testicular homogenate compared to the control group. Chemotherapy drugs cause mitochondrial damage, leading to increased reactive oxygen species (ROS) and oxidative stress [56–58]. This ROS accumulation acts as a chemoattractant, attracting MSCs and EVs to the damage site [59]. Consistent with earlier findings cis increased MDA levels, a metabolit of lipid peroxidation, and significantly raised TNF-α, an inflammatory marker [2,3,10,24,25,36]. The EVs-treated Cis group showed significant improvements in GSH-PX, SOD, and CAT activities and a significant reduction in MDA levels. Although single-dose EVs reduced TNF-α levels, the decrease was not significant. Elevated TNF-α levels impair spermatogenesis and reduce testosterone levels, with increased ROS production contributing to this process through apoptosis [60–63]. Studies have demonstrated that EVs have nano-particle size can penetrate bodily barriers to reach damaged areas [34,64,65]. EVs with anti-apoptotic, anticancer, and antioxidant properties have been shown to reduce oxidative stress at the damage site [17,54,65,66].

The study's reliance on a single dose of EVs and the subsequent evaluations conducted over a short follow-up period inherently limit the ability to assess long-term effects and the sustainability of treatment. Exploring the efficacy of EVs administered at varying doses and through different delivery methods could provide valuable insights into the optimal therapeutic approach. Additionally, expanding the scope of the investigation by incorporating additional parameters such as sperm quality assessments, detailed testicular histopathology, and comprehensive biomarker analyses would facilitate a more nuanced understanding of the underlying mechanisms of action associated with EVs treatment.

## Conclusion

EVs offer numerous advantages as a cell-free therapy and have shown potential in treating various human diseases in recent years. They are natural, can be derived from a person's own cells, and targeted to specific cells or tissues. This study investigated the therapeutic efficacy of bone marrow-derived MSC EVs on cis-induced testicular damage in rats. EVs, possessing regenerative and anti-cancer properties, were found to mitigate cis-induced testicular damage, suggesting their potential as a treatment for such damage and germ cell layer degeneration. This study indicates that a single dose of EVs can ameliorate Cis-induced testicular damage by reducing oxidative stress and regulating autophagy. Further studies should explore repeated doses, different dosing regimens, and more localized applications for more effective and long-term results. Additionally, research focusing on the molecular regulation of autophagy may provide a deeper understanding of its therapeutic effects.

## Acknowledgment

We thank the staff of the Erciyes University ERU Experimental Research and Application Centre for their care of the rats used in this study. We also appreciate the support from the Gevher Nesibe Genome and Stem Cell Institute (GENKÖK) of Erciyes University in generating BM-MSCs and the EVs derived from these cells.

## Author contributions

**Conceptualization:** Halime Tozak Yıldız, Zeynep Burçin Gönen, Arzu Yay.

**Data curation:** Halime Tozak Yıldız, Kübra Tuğçe Kalkan.

**Formal analysis:** Kübra Tuğçe Kalkan, Özge Cengiz Mat, Eda Köseoğlu.

**Investigation:** Gözde Özge Önder.

**Methodology:** Kübra Tuğçe Kalkan, Numan Baydilli, Zeynep Burçin Gönen, Özge Cengiz Mat, Eda Köseoğlu.

**Project administration:** Gözde Özge Önder.

**Resources:** Halime Tozak Yıldız.

**Software:** Kübra Tuğçe Kalkan, Eda Köseoğlu.

**Supervision:** Numan Baydilli, Özge Cengiz Mat, Arzu Yay.

**Validation:** Numan Baydilli, Zeynep Burçin Gönen, Özge Cengiz Mat, Gözde Özge Önder.

**Visualization:** Arzu Yay.

**Writing – original draft:** Halime Tozak Yıldız.

**Writing – review & editing:** Halime Tozak Yıldız, Arzu Yay.

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
