## [Decision Letter · Decision Letter 0]

4 Sep 2024

PONE-D-24-32691EVs Therapy for Cisplatin-Induced Testicular Tissue ToxicityPLOS ONE

Dear Dr. TOZAK YILDIZ,

Thank you for submitting your manuscript to PLOS ONE. After careful consideration, we feel that it has merit but does not fully meet PLOS ONE’s publication criteria as it currently stands. Therefore, we invite you to submit a revised version of the manuscript that addresses the points raised during the review process.

We look forward to receiving your revised manuscript.

Kind regards,

Roland Eghoghosoa Akhigbe

Academic Editor

PLOS ONE

Journal Requirements:

2. We note that your Data Availability Statement is currently as follows: 

"All relevant data are within the manuscript and its Supporting Information files."

**Additional Editor Comments:**

The study by Halime et al. is interesting and demonstrates the therapeutic value of EVs in attenuating cisplatin-induced testicular toxicity. However, just like both reviewers have suggested a major revision, the manuscript needs a thorough revision to meet the standard of this journal. Authors should also consider editing the manuscript for grammar and syntax. In addition to the reviewers' comments, these comments would be useful:

Abstract

1. What are the metabolites of cisplatin that exert its effect

2. “MDA activity”. Is MDA an enzyme?

3. Include recommendations

Introduction

1. Paragraph 1: The effects should precede the mechanisms

2. Paragraph 3: The note on autophagy is hanging. It is not linked with cisplatin nor EV

Methods/Discussion

1. The immunolocalization of the assayed molecules should be discussed rather than the expression only

2. Since cisplatin also suppresses hypothalamic-pituitary-testicular axis, it is important to evaluate gonadotropins and GnRH

3. How long did the study last?

4. Was spermatogenic cycle studied?

5. Why is the histology of the epididymis not presented?

6. It is not sufficient to report sperm count and motility and ignore other sperm physiological functions. Are others not relevant?

Reviewers' comments:

Reviewer's Responses to Questions

**Comments to the Author**

1. Is the manuscript technically sound, and do the data support the conclusions?

Reviewer #1: Yes

Reviewer #2: Yes

2. Has the statistical analysis been performed appropriately and rigorously? 

Reviewer #1: Yes

Reviewer #2: Yes

3. Have the authors made all data underlying the findings in their manuscript fully available?

Reviewer #1: Yes

Reviewer #2: Yes

4. Is the manuscript presented in an intelligible fashion and written in standard English?

Reviewer #1: No

Reviewer #2: No

5. Review Comments to the Author

Reviewer #1: Although this study is interesting, there are many problems with this manuscript, mainly due to missing information in the text, improperly explained facts, and missing references. For all the details, I attach a PDF with all reviewer comments.

Reviewer #2: Hi Dr Yildiz

I have gone through your work and want to commend you for the effort which you have made to put this together, especially seeing that this project is likely self-funded. However, I have the following comments to make this better for the entire scientific community;

1. The use of English in this draft does not represent the best scientific expertise expected for a paper like this.

2. From the opening sentence of your abstract , you failed to ignite my enthusiasm about this project. The significance of cisplatin...why cisplatin? Justification for the study? The purpose section in your abstract must be rewritten. This manuscript have great potential if you can do that. Then, the methods... Rewrite your methods so we understand your methodology well...length of experiment, animals? How many? You left me clueless about this experiment here! Then, how did you get your data? Laboratory analysis? Statistical tests? You said testicular morphology was examined using H&E?? Really???? Histoarchitecture or morphology ?? Please add the methods you used for the measurement of hormones, sperm parameters. Your methodology is very poor in this abstract in that... You omitted so many details....

3. The results subsection of your abstract is also faultable. I expect that the results will be presented in a logical sense that follows a thorough discussion of effect and mechanism.

For example; first state the effects on structure, Hormone (function), sperm parameters (function) then the mechanisms of these negative or positive changes (oxidative stress, autophagy)

4. The conclusion of the abstract does not show comprehensive snapshot of your results. I guess you were have a wordcount limit for your abstract but you have to do these corrections to put this manuscript in top shape.

5. Your introduction is needs to be overhauled. From the first sentence with reference (Cisplatin I'd a potent....) to " it has been reported" when you actually didn't add those studies that reported. The third sentence on the genotoxic effects of Cis also lacks merit? Formation of primordial follicle is a genotoxic effects?? Are you sure? You indicated that Cis reduces spermatogenesis? How? What sperm parameters? What stage of spermatogenesis?

6. Then you described EVs as if you didn't get the description somewhere, why? This dampens my enthusiasm, honestly! Each factual statement you make should be backed by a reference.

7. The next paragraph is same. So many missing references. Each factual statement should end with a reference.

8. Replace "long-lived" with a more appropriate adjective.

9. I see no need for the "much research has been focused on starvation-induced autophagy..." What are you trying to say? A lot of missing connections in the paragraph.

10. These paragraphs in the introduction do not connect.

11. I was expecting your introduction will shadow this line of thought

Few sentences on general info about cisplatin. Then it's toxicity on general physiology in humans and animal models. Then, the reprotoxic effects. This should be detailed and more of males. After stating the effects, you discuss the mechanisms. Then, the following paragraph will tow...

"On the other hand, EVs are bla bla..." After the general info about EVs you should talk about effects in several reprotoxicity models , preferably cisplatin or anticancer drugs (if available). You should discuss the mechanisms of its effects exhaustively.

11. Be sure to cite similar works https://doi.org/10.1007/s43188-024-00250-3 in your methodology especially for spermatogenesis, hormones, oxidative stress.

12. Your discussion should tow the same line for the presentation of result. Always be sure to first summarize your findings in the first paragraph of discussion, then discuss histology and morphological parameters in the next paragraph. The following paragraph should discuss findings on hormones...

13. Next, spermatogenesis and sperm parameters, then go to the mechanisms in the subsequent paragraph.

14. Your discussion should include strengths and weaknesses of this study. Your methodology is unique and I am sure there are good things and limitations to it.

15. If possible include graphical abstract. This is not compulsory though.

6. PLOS authors have the option to publish the peer review history of their article (what does this mean? ). If published, this will include your full peer review and any attached files.

**Do you want your identity to be public for this peer review?** For information about this choice, including consent withdrawal, please see our Privacy Policy .

Reviewer #1: No

Reviewer #2: **Yes: ** Oyedokun PRECIOUS ADEOYE

---

## [Author Response · Author response to Decision Letter 0]

16 Oct 2024

Dr.Halime TOZAK YILDIZ (Ph.D.)

Kırşehir Ahi Evran University Medical Faculty

Department of Histology and Embryology, Campus 40100 Kirsehir, Turkey

Erciyes University Institute of Healty Science, Stem Cell Sciences, Doctorate Student

E-mail adress: htyildiz@ahievran.edu.tr, hhalimeyildiz@hotmail.com

Dear Roland Eghoghosoa Akhigbe

We would like to thank you for writing on 05/09/2024 and for the opportunity to resubmit a revised copy of this manuscript. At this point we would also like to thank the reviewers for the positive feedback and the helpful comments on corrections or changes.

We believe this has resulted in an improved revised manuscript, which you will find uploaded alongside this document. The manuscript has been revised to reflect the reviewer's comments accompanying our replies to this letter. We sincerely hope that the revised manuscript will be accepted for publication in the Plos One.

Please address all correspondence concerning this manuscript to me at htyildiz@ahievran.edu.tr

Thank you for your consideration of this manuscript.

Yours sincerely

Dr.Halime TOZAK YILDIZ (PhD)

Editor Comments

Comment_1 comments on the abstract;

1. What are the metabolites of cisplatin that exert its effect

2. “MDA activity”. Is MDA an enzyme?

3. Include recommendations

Response_1

Many thanks to our referee for his time and valuable comment. According to the reviewers' comments, the revised version of the text is given in the article.

1. Because of the word limit in the abstract, a correction was made in the second paragraph of the introduction.

2. MDA is a metabolite released as a result of lipid peroxidation. Corrections have been made in the relevant parts of the text.

3. In the abstract, a suggestion sentence has been added to the conclusion section.

“Further exploration of dosing regimens and localized applications is recommended for improved efficacy.”

Comment_2 comments on the introduction;

1. Paragraph 1: The effects should precede the mechanisms

2. Paragraph 3: The note on autophagy is hanging. It is not linked with cisplatin nor EV

Response_2

Many thanks to our referee for his insightful comment. According to the reviewers' comments, the redacted version of the text is given in the article (green text highlighting indicates revised texts)

1. the first two paragraphs have been corrected.

“Cisplatin (Cis) is a widely used chemotherapy drug for treating various types of cancer, including small cell lung, testicular, ovarian, and breast cancers. It functions primarily by forming DNA cross-links that inhibit DNA replication, leading to cell cycle arrest, apoptosis and autophagy in rapidly dividing cancer cells [1]. However, Cisplatin’s use is limited by its significant side effects, including its toxicity to the reproductive system [2-10].

Cisplatin's intracellular metabolites are primarily known as aquated metabolites (such as mono/di-aqua cisplatin). Once inside the cell, cisplatin undergoes hydrolysis, losing chloride ions and forming its active species [5]. The mechanism of Cisplatin-induced reproductive toxicity is closely associated with the production of reactive oxygen species (ROS) during its metabolism, which leads to oxidative stress and subsequent tissue damage [2, 3, 5, 8-10]. In females, Cisplatin has been shown to cause apoptosis in granulosa cells, which support the development of oocytes, leading to a reduction in ovarian reserve and impaired folliculogenesis [6]. In males, Cisplatin negatively impacts testicular function, affecting Leydig cells, seminiferous tubules, spermatogenic cells, and Sertoli cells, which are crucial for maintaining normal spermatogenesis [8]. This damage manifests as reduced sperm count, altered sperm morphology, impaired chromatin integrity, and decreased sperm motility, ultimately contributing to infertility [4-12]. The disruption of spermatogenesis by Cisplatin involves damage at multiple stages, including spermatogonia, spermatocytes, and spermatids, leading to impaired sperm development and maturation [12].”

2. Autophagy has been linked to cisplatin.

“Notably, autophagy can be stimulated by anti-cancer drugs, which promote the degradation of tumor cells; however, while some oncoproteins inhibit autophagy, many tumor suppressor genes enhance it [20].”

Comment_3 comments on the methods/discussion;

1. The immunolocalization of the assayed molecules should be discussed rather than the expression only

2. Since cisplatin also suppresses hypothalamic-pituitary-testicular axis, it is important to evaluate gonadotropins and GnRH

3. How long did the study last?

4. Was spermatogenic cycle studied?

5. Why is the histology of the epididymis not presented?

6. It is not sufficient to report sperm count and motility and ignore other sperm physiological functions. Are others not relevant?

Response_3

Many thanks to our referee for his valuable comment. According to the reviewers' comments, the redacted version of the text is given in the article. Green text highlighting indicates revised texts, yellow text highlighting indicates new texts,

1. Immunolocalisation of molecules has been added to the related paragraph

2. The project could not be funded, so testicular hormones of the hypothalamic-pituitary-testicular axis had to be preferred.

3. The study lasted 8 days.

4. The spermatogenic cycle was not analysed because Johnsen testicular biopsy score was evaluated.

5. Epidididymis dissected only to obtain stored spermatozoa, histology could not be analysed.

6. We thank the reviewers for emphasizing the limitations of evaluating sperm function based on sperm count and sperm motility alone. We agree that a more comprehensive physiological assessment of sperm is important to understand sperm quality more completely. This is especially true when examining the effects of chemotherapy-induced testicular damage. In this study, we examined markers of oxidative stress (e.g. malondialdehyde [MDA], glutathione peroxidase [GSH-PX], superoxide dismutase [SOD] and catalase [CAT]) and autophagy (Beclin-1, p62, LC3-2) to more precisely assess sperm health. It has been shown in the literature that both can have a negative impact on sperm production and quality. These processes damage sperm DNA, lipids, and proteins, resulting in reduced sperm motility and increased DNA fragmentation (En et al., 2020; Abdel‐latif et al., 2022). These additional evaluations provide a clearer understanding of the mechanisms affecting sperm quality and allow for a more complete assessment of sperm health after cisplatin-induced testicular damage.

References

1. En, L. T., Brougham, M., Wallace, W. H., & Mitchell, R. T. (2020). Impacts of platinum-based chemotherapy on subsequent testicular function and fertility in boys with cancer. Human Reproduction Update, 26(6), 874-885. https://doi.org/10.1093/humupd/dmaa041.

2. Abdel‐latif, R. G., Fathy, M., Anwar, H. A., Naseem, M., Dandekar, T., & Othman, E. M. (2022). Cisplatin-induced reproductive toxicity and oxidative stress: ameliorative effect of kinetin. Antioxidants, 11(5), 863. https://doi.org/10.3390/antiox11050863.

I hope these adjustments better communicate the significance of our research and improve the overall flow of the abstract. Your input has been invaluable in this process, and I am grateful for the opportunity to refine our work.

Reviewer 1

Comment_1

No abbreviations in the title. Please write this out. Title should provide information in the model used..

Response_1

Many thanks to our referee for his time and valuable comment. According to the reviewers' comments, the revised version of the text is given in the article.

Comment_2 comments on the abstract;

- It must firs be said which model was used?

- How many in each group?

-Indicate whether these changes are significant, with P-values)

Response_2

Many thanks to our referee for his valuable comment. According to the reviewers' comments, the redacted version of the text is given in the article (green text highlighting indicates revised texts)

Comment_3 comments on the introduction

-References

-Word mistakes and repetitions

-Disorders of meaning

Response_3

Many thanks to our referee for his valuable comment. According to the reviewers' comments, the redacted version of the text is given in the article. Green text highlighting indicates revised texts, yellow text highlighting indicates new texts,

Comment_4 comments on the methods

-Supplier, city, country?

- Word mistakes

- Missing information on the method

Response_4

Many thanks to our referee for his valuable comment. According to the reviewers' comments, the article has been edited and supplier information has been added. Green text highlighting indicates revised texts

Comment_5 comments on the discussion

-Rephrase for better reading. Four subsequent sentences start with the same word.

- İncomplete and unclearly sentences

-References

Response_5

Many thanks to our referee for his valuable comment. According to the reviewers' comments, the redacted version of the text is given in the article. Green text highlighting indicates revised texts, yellow text highlighting indicates new texts,

I hope these adjustments better communicate the significance of our research and improve the overall flow of the abstract. Your input has been invaluable in this process, and I am grateful for the opportunity to refine our work.

Reviewer 2

Comment_1

The use of English in this draft does not represent the best scientific expertise expected for a paper like this.

Response_1

We would like to thank the referee for his valuable comments. The text has been revised and corrected according to the comments of the referees.

Comment_2

From the opening sentence of your abstract , you failed to ignite my enthusiasm about this project. The significance of cisplatin...why cisplatin? Justification for the study? The purpose section in your abstract must be rewritten. This manuscript have great potential if you can do that. Then, the methods... Rewrite your methods so we understand your methodology well...length of experiment, animals? How many? You left me clueless about this experiment here! Then, how did you get your data? Laboratory analysis? Statistical tests? You said testicular morphology was examined using H&E?? Really???? Histoarchitecture or morphology ?? Please add the methods you used for the measurement of hormones, sperm parameters. Your methodology is very poor in this abstract in that... You omitted so many details....

Response_2

Many thanks to our referee for his insightful comment. I appreciate your suggestions regarding the clarity and organization of the abstract. In response, I have revised the structure and content to enhance readability and conciseness.

As you suggested, I have restructured the headings within the abstract to provide clearer delineation of key points. The previous sections, which are now marked through, have been replaced with newly emphasized green text that aims to present our findings in a more straightforward manner. I hope these adjustments better communicate the significance of our research and improve the overall flow of the abstract.

Comment_3

The results subsection of your abstract is also faultable. I expect that the results will be presented in a logical sense that follows a thorough discussion of effect and mechanism.

For example; first state the effects on structure, Hormone (function), sperm parameters (function) then the mechanisms of these negative or positive changes (oxidative stress, autophagy)

Response_3

We agree with the reviewer’s assessment of the analysis. The editing requested by the reviewer was made in the article.

“Cisplatin exposure led to significant testicular damage, evidenced by reduced germinal epithelium and seminiferous tubule degeneration (p<0.001). These structural alterations resulted in hormonal declines, notably in testosterone (p<0.005) and inhibin (p<0.001), along with decreased sperm motility (p<0.05) and count (p<0.001). Immunohistochemical analysis revealed upregulated autophagy markers (p<0.001) and downregulated steroidogenic factors (p<0.001), indicating disrupted steroidogenesis. Increased malondialdehyde (MDA) levels (p<0.01) and reduced antioxidant enzyme activities (p<0.001) highlighted oxidative stress. Conversely, EV treatment improved histoarchitecture (p<0.001), restored hormone levels (testosterone p<0.005, inhibin p<0.001), regulated the expression of autophagy markers (p<0.001) and enhanced antioxidant activities (p<0.001), demonstrating a protective effect.”

Comment_4

The conclusion of the abstract does not show comprehensive snapshot of your results. I guess you were have a wordcount limit for your abstract but you have to do these corrections to put this manuscript in top shape.

Response_4

Many thanks to our referee for his insightful comment. As you rightly pointed out, the word limit in the abstract may prevent certain definitions from being fully articulated. I have made the necessary amendments in accordance with your recommendations.

“Conclusion: EVs protect against cisplatin-induced reproductive toxicity by modulating oxidative stress and autophagy pathways, preserving testicular function and fertility. These findings suggest that EVs may be a promising therapeutic strategy for mitigating cisplatin's negative effects on reproductive health. Further exploration of dosing regimens and localized applications is recommended for improved efficacy”

Comment_5

Your introduction is needs to be overhauled. From the first sentence with reference (Cisplatin I'd a potent....) to " it has been reported" when you actually didn't add those studies that reported. The third sentence on the genotoxic effects of Cis also lacks merit? Formation of primordial follicle is a genotoxic effects? Are you sure? You indicated that Cis reduces spermatogenesis? How? What sperm parameters? What stage of spermatogenesis?

Response_5

Many thanks to our referee for his time and valuable comment. According to the reviewers' comments, the redacted version of the text is given in the article. Green text highlighting indicates revised texts, yellow text highlighting indicates new texts,

Comment_6

Then you described EVs as if you didn't get the description somewhere, why? This dampens my enthusiasm, honestly! Each factual statement you make should be backed by a reference.

Response_6

Many thanks to our referee for his valuable comment. The relevant references have been incorporated into the sentence defining EVs. Additionally, the entire paragraph has been carefully reviewed and revised.

Comment_7

The next paragraph is same. So many missing references. Each factual statement should end with a reference.

Response_7

I would like to thanks to the referee for his valuable comment. The references have been thoroughly reviewed and appropriately added to the relevant sentences.

Comment_8

Replace "long-lived" with a more appropriate adjective.

Response_8

Instead of the expression "long-lived", the expression “persistent” is used as a more appropriate expression.

Comment_9

I see no need for the "much research has been focused on starvation-induced autophagy..." What are you trying to say? A lot of missing connections in the paragraph.

Response_9

Many thanks to our referee for his valuable comment. he entire paragraph has been reviewed and revised in accordance with the critiques provide. The unnecessary expression has been removed.

Comment_10

These paragraphs in the introduction do not connect.

I was expecting your introduction will shadow this line of thought.

Few sentences on general info about cisplatin. Then it's toxicity on general physiology in humans and animal models. Then, the reprotoxic effects. This should be detailed and more of males. After stating the effects, you discuss the mechanisms. Then, the following paragraph will tow...

"On the other hand, EVs are bla bla..." After the general info about EVs you should talk about effects in several reprotoxicity models , preferably cisplatin or anticancer drugs (if available). You should discuss the mechanisms of its effects exhaustively.

Response_10

I would like to thanks to the referee for his valuable comment. In accordance with your feedback, t

---

## [Decision Letter · Decision Letter 1]

6 Nov 2024

Extracellular Vesicles Therapy Alleviates Cisplatin-Induced Testicular Tissue Toxicity in a Rat Model

PONE-D-24-32691R1

Dear Dr. TOZAK YILDIZ,

We’re pleased to inform you that your manuscript has been judged scientifically suitable for publication and will be formally accepted for publication once it meets all outstanding technical requirements.

Kind regards,

Roland Eghoghosoa Akhigbe

Academic Editor

PLOS ONE

Additional Editor Comments (optional):

Reviewers' comments:

Reviewer's Responses to Questions

**Comments to the Author**

1. If the authors have adequately addressed your comments raised in a previous round of review and you feel that this manuscript is now acceptable for publication, you may indicate that here to bypass the “Comments to the Author” section, enter your conflict of interest statement in the “Confidential to Editor” section, and submit your "Accept" recommendation.

Reviewer #1: All comments have been addressed

Reviewer #2: All comments have been addressed

2. Is the manuscript technically sound, and do the data support the conclusions?

Reviewer #1: Yes

Reviewer #2: Yes

3. Has the statistical analysis been performed appropriately and rigorously? 

Reviewer #1: Yes

Reviewer #2: Yes

4. Have the authors made all data underlying the findings in their manuscript fully available?

Reviewer #1: Yes

Reviewer #2: Yes

5. Is the manuscript presented in an intelligible fashion and written in standard English?

Reviewer #1: Yes

Reviewer #2: Yes

6. Review Comments to the Author

Reviewer #1: The authors have basically reworked their submission and thereby have improved it significantly. Except for a few minor editorial corrections that should be done, I am satisfied. I attach a PDF in which I indicated these issues.

Reviewer #2: The authors have addressed all comments previously raised. I am satisfied with the current version of the paper.

7. PLOS authors have the option to publish the peer review history of their article (what does this mean? ). If published, this will include your full peer review and any attached files.

**Do you want your identity to be public for this peer review?** For information about this choice, including consent withdrawal, please see our Privacy Policy .

Reviewer #1: No

Reviewer #2: **Yes: ** Oyedokun Precious Adeoye

---

## [Editor Report · Acceptance letter]

PONE-D-24-32691R1

PLOS ONE

Dear Dr. Tozak Yıldız,

I'm pleased to inform you that your manuscript has been deemed suitable for publication in PLOS ONE. Congratulations! Your manuscript is now being handed over to our production team.

Kind regards,

on behalf of

Dr. Roland Eghoghosoa Akhigbe

Academic Editor

PLOS ONE